# Bee Pollen as a Dietary Mineral Source: In Vitro Bioaccessibility and Health Risk Evaluation

**DOI:** 10.3390/molecules30132745

**Published:** 2025-06-26

**Authors:** Ceren Mutlu, Volkan Aylanc, Miguel Vilas-Boas

**Affiliations:** 1CIMO, LA SusTEC, Instituto Politécnico de Bragança, Campus de Santa Apolónia, 5300-253 Bragança, Portugal; volkanaylanc@yahoo.com; 2Food Engineering Department, Engineering Faculty, Balıkesir University, 10145 Balıkesir, Türkiye; 3LAQV-REQUIMTE, Departamento de Química e Bioquímica, Faculdade de Ciências, Universidade do Porto, 4169-007 Porto, Portugal

**Keywords:** micronutrients, bioaccessibility, recommended daily allowance, target hazard quotient

## Abstract

Pollen, the male gametophyte of flowering plants, is collected by honeybees as a primary source of protein and converted into bee pollen through the enzymatic activity of digestive secretions. The nutrients in bee pollen are available in amounts well beyond those of proteins, comprising macronutrients such as carbohydrates, lipids and dietary fiber, as well as micronutrients such as minerals, vitamins, organic acids, and phenolic compounds. This study aimed to determine the macro and trace mineral content of bee pollen from different botanical and geographical origins, and to assess their bioaccessibility through simulated in vitro digestion, their dietary contribution, and potential health risks. Seven bee pollen samples were investigated, three with a monofloral origin of above 80%, from *Nigella* spp., *Helianthus annuus* and *Castanea sativa*, and four with a multifloral origin. Mineral composition revealed potassium as the most abundant element, while iron, manganese, and copper were found at trace levels. *Castanea sativa* pollen had the highest overall mineral content, whereas *Nigella* spp. showed the lowest values for calcium, magnesium, and copper. The bioaccessibility of bee pollen was highest during the gastric phase for most minerals except copper, where most of the samples peaked in the intestinal phase. Overall, mineral bioaccessibility after simulated digestion followed the order K > Mg > Cu > Ca > Mn > Fe > Zn. While for manganese, the consumption of bee pollen showed the highest contribution to recommended dietary intake (16% for women and 12% for men), calcium had the lowest, with less than 1% of the RDA at a consumption level of 40 g/day. Health risk assessment confirmed that consuming 40 g/day of bee pollen poses no risk because the target hazard quotient and hazard index are below the risk threshold of 1.0.

## 1. Introduction

Pollen, the male gametophyte of flowering plants, is collected by honeybees as a primary source of protein and converted into bee pollen through the enzymatic activity of digestive secretions, along with the addition of nectar and bee-derived substances. Besides its high protein content (10–40%), bee pollen comprises a wide range of macronutrients including carbohydrates (13–55%), lipids (1–12%), and dietary fiber (0.3–20%), as well as micronutrients such as minerals (0.2–3.0%), vitamins (C, E, and K), organic acids, and phenolic compounds [1,2,3,4]. Among these constituents, minerals are of particular significance due to both their nutritional value and environmental relevance [5]. Bee pollen is recognized as a rich source of essential minerals, notably potassium (K), phosphorus (P), calcium (Ca), magnesium (Mg), iron (Fe), copper (Cu), and zinc (Zn). It also contains trace elements such as manganese (Mn), molybdenum (Mo), nickel (Ni), chromium (Cr), selenium (Se), and silicon (Si) [1,4,5,6]. These minerals are vital for both the development of honeybee colonies and for human health—particularly Ca, Fe, Mg, Mn, and Zn—due to their indispensable roles in enzymatic activity, cellular protection, homeostasis, and various metabolic processes [6,7].

As consumer awareness of food quality and safety continues to grow, determining the mineral composition of food products has become increasingly important for evaluating both their nutritional value and potential environmental contamination. Consequently, studies focused on the mineral composition of bee pollen using various analytical techniques have become more widespread [1,2,3,4,6,8]. Among these techniques, atomic absorption spectroscopy (AAS) stands out as a relatively inexpensive, user-friendly, and widely used method for detecting trace and ultra-trace levels of minerals [9,10]. It has been extensively applied to assess the mineral content of a wide range of food products, including grape juice, wine, dairy products, various cereals, and seeds such as rice, chia, amaranth, quinoa, sunflower, and pumpkin [10].

Although substantial data exist regarding the nutritional composition of bee pollen, a comprehensive assessment of its nutritional potential must go beyond total mineral content. It must also account for the bioaccessibility of individual nutrients during gastrointestinal digestion. Bioaccessibility refers to the fraction of a nutrient, such as a mineral, that is released from the food matrix during gastrointestinal digestion and is potentially available for absorption through the intestinal epithelium into the systemic circulation [11]. To assess bioaccessibility, researchers employ both in vitro and in vivo methodologies. Among these, in vitro techniques are explored due to their simplicity, lack of ethical constraints, and ability to simulate gastrointestinal conditions using standardized protocols such as the INFOGEST method developed by an international consortium focused on food digestion [12]. Previous studies have investigated the in vitro digestibility of carbohydrates (e.g., fructose, glucose, sucrose, maltose, and maltulose), proteins [1], and phenolic compounds [13,14] in bee pollen, but none have investigated that in minerals.

While numerous studies have explored the mineral content of bee pollen [1,2,3,4,6,8], no prior study has provided a thorough evaluation of mineral bioaccessibility during in vitro digestion and its implications for human health. Therefore, the main objectives of this study were to determine the concentrations of macro- and trace minerals (K, Ca, Mg, Fe, Mn, Cu, and Zn) in bee pollen samples with diverse botanical and geographical origins, evaluate their bioaccessibility through simulated in vitro digestion stages using the INFOGEST procedure, assess their potential contribution to the daily dietary allowance, and estimate associated health risks for adult women and men.

## 2. Results and Discussion

### 2.1. Botanical Origins and Mineral Composition of Undigested Bee Pollens

Bee pollen samples containing 80% or more of a single pollen type were classified as monofloral, while the remaining samples were categorized as multifloral, in accordance with the ISO/FDIS 24382 standard [15]. Based on this classification, samples BP1, BP2, BP4, and BP6 were identified as multifloral (Table 1). In contrast, BP3 consisted predominantly of *Nigella* spp. (85%), BP5 of *Helianthus annuus* (90%), and BP7 of *Castanea sativa* (83%); these were therefore classified as monofloral.

The concentrations of K, Ca, Mg, Fe, Mn, Cu, and Zn in both monofloral and multifloral bee pollen samples, determined using AAS, are summarized in Table 2, and their average values in undigested bee pollen are shown in Figure 1. Potassium was the most abundant mineral, with concentrations ranging from 2420 to 6854 mg/kg, while Ca, Mg, and Zn appear at a second level with average concentrations of 475 mg/kg, 624 mg/kg, and 120 mg/kg, respectively. The elements Fe, Mn, and Cu were present at levels below 60 mg/kg. Among all samples, *Castanea sativa* bee pollen exhibited a significantly higher concentrations of most minerals (*p* < 0.01), with the exception of Mg, which was more prevalent in multifloral samples (BP1, BP4, and BP6) than in monofloral ones. Conversely, the lowest levels of Ca, Mg, and Cu were detected in *Nigella* spp. bee pollen.

Previous studies have consistently demonstrated that K, P, Mg, Ca, and Na are the predominant minerals in bee pollen, based on their concentrations [6,8,16,17,18]. According to results reported by Çobanoğlu et al. [17], K alone accounts for approximately 77% of the total mineral content, while Mg, Na, and Ca collectively represent around 15% in *Verbascum* spp. and some multifloral bee pollen samples. Potassium is widely recognized as the most abundant element, with reported concentrations ranging from 485 to 10,340 mg/kg [8,16,17,19]. On the other hand, Ca and Mg contents have been reported in the range of 85–2202 mg/kg and 503–2150 mg/kg, respectively [4,8,17]. In contrast, Fe, Mn, Cu, and Zn are generally classified as trace elements in bee pollen and are present in lower concentrations, consistent with the findings of this study [6,16,17]. Reported levels for these trace minerals range from 17 to 811 mg/kg for Fe, 14–92 mg/kg for Mn, 2–26 mg/kg for Cu, and 15–82 mg/kg for Zn [4,8,16,17].

Variations in the mineral composition and content of bee pollen have been linked to a range of environmental and biological factors. It has been mentioned that mineral levels in bee pollen are influenced by plant nutrient uptake, the mineral composition of the soil in which the plants grow, and seasonal rainfall patterns [6]. Furthermore, changes in floral diversity throughout the year contribute to fluctuations in mineral concentrations, with bees tending to collect pollen from Na-rich plants during summer and from K-rich sources during winter. Kostić et al. [8] reported that minerals such as K, Ca, Mg, and Mn, which are absorbed by plants via their roots, are likely to reflect the mineral characteristics of the soil. However, the mineral content of bee pollen does not always show a direct correlation with geographical origin, suggesting that other factors—such as mineral uptake from air and water—may also play a role, with absorption capacities varying among different plant species. Similarly, Valverde et al. [6] emphasized that the observed differences in mineral content are more strongly associated with the botanical origin of the pollen/harvest period rather than the geographical location of the apiary.

The mineral compositions of *Castanea sativa* [2], *Helianthus annus*, and *Salix* spp. have been revealed in previous studies [8,20]. Kolayli et al. [2] found that *Castanea sativa* bee pollen contained 5822 mg/kg of K, 1485 mg/kg of Ca, 569 mg/kg of Mg, 152 mg/kg of Fe, 113 mg/kg of Mn, 56 mg/kg of Zn, and 13 mg/kg of Cu. In comparison, the present study revealed higher levels of K, Mg, Fe, and Zn, while the Ca concentration was approximately 1.8 times lower than that in the reported results by Kolayli et al. [2]. Interestingly, both studies consistently identified significantly higher Mn levels in *Castanea sativa* bee pollen compared to other monofloral or multifloral samples. The variations may be attributed to differences in geographical and climatic conditions. The *Castanea sativa* sample in this study was collected from the northwestern region of Türkiye (Yalova), characterized by higher industrial density and relatively lower rainfall, whereas the samples used in the previously reported study were sourced from the northeastern region (Artvin-Rize), which is less industrialized and receives more precipitation. It has been suggested that industrial activity may elevate mineral content, while high rainfall can leach minerals from the soil, reducing their availability for plant uptake [6].

Besides *Castanea sativa*, Kostić et al. [8] also analyzed bee pollen from *Helianthus* and *Salix* species. Comparatively, they reported that *Helianthus* pollen had high Fe content (141 mg/kg), whereas *Salix* pollen contained higher levels of K (3404 mg/kg), Mg (595 mg/kg), and Mn (23 mg/kg). Ca, Zn, and Cu levels in both pollen types were relatively similar, ranging between 1189 and 1196 mg/kg for Ca, 41 and 45 mg/kg for Zn, and 6.3- and 0.6 mg/kg for Cu. The findings of the present study indicated that the Zn contents of *Helianthus annuus* bee pollen and BP2, rich in *Salix* spp., were comparable, while the Mg and Mn concentrations were significantly higher in BP2, but Fe content was larger in *Helianthus annuus* bee pollen, consistent with the results reported by Kostić et al. [8]. In another study, the contents of K, Ca, Mg, and Fe were found to be notably higher in *Salix* spp. bee pollen compared to those in *Helianthus annuus* L. bee pollen, whereas Zn content showed no significant difference [20].

### 2.2. Mineral Composition Changes During In Vitro Digestion and Bioaccessibility Results

While the total mineral content of a food matrix offers valuable compositional insights, it does not necessarily reflect the proportion that is bioaccessible for absorption in the human body. Therefore, to gain a clear value of bee pollen as a source of minerals for human nutrition, the in vitro gastrointestinal digestion model was performed, assessing the bioaccessibility of minerals in these bee products. The changes in mineral concentrations across different digestion stages are presented in Table 2, while the bioaccessibility percentages for each mineral are illustrated in Figure 2.

The results of mineral composition analysis of samples taken from the oral, gastric, and intestinal digestion stages revealed that, for most minerals, the gastric phase represents the highest release, with the exception of Cu and three samples for Fe. Indeed, for copper, the highest bioaccessibility was observed in the intestinal phase, which is particularly significant for bee pollen of *Nigella* spp. and *Helianthus annuus*, where the increase from the gastric phase represents more than 10% (Figure 2F). Following gastric digestion, a marked reduction in the levels of K, Ca, Mg, Mn, and Zn was observed in all bee pollen samples.

Although specific studies on the mineral changes in bee pollen during in vitro digestion are lacking, similar trends have been reported in other plant-based foods. Scrob et al. [21] found that in dried fruits such as dates, coconut, cranberries, prunes, and bananas, minerals like K, Mg, Ca, Fe, Mn, and Zn were most bioaccessible during the gastric phase—consistent with the findings of the present study. The increased mineral solubility in this phase is attributed to the acidic environment of the stomach. Conversely, during the intestinal phase, factors such as mineral precipitation, complex formation, or interactions with other components may reduce bioaccessibility.

After simulated digestion, the mineral bioaccessibility in bee pollen samples followed the order K > Mg > Cu > Ca > Mn > Fe > Zn. Notably, potassium concentrations were substantially higher than those in undigested samples, as shown in Figure 2A. The mean bioaccessibility values for K were 158%, 230%, and 184% in the oral, gastric, and intestinal phases, respectively. Among the bee pollen samples, the *Castanea sativa* BP7 sample exhibited the lowest K bioaccessibility, while the *Salix* spp.-rich bee pollen of sample BP2 demonstrated the highest bioaccessibility at the end of digestion. The bioaccessibility of K is primarily influenced by other nutrients present in food, such as proteins, amino acids, carbohydrates, and various minerals, which can form both soluble and insoluble complexes with potassium. Additionally, anti-nutritional factors like phytic acid, oxalates, and tannins promote the formation of these complexes and can significantly inhibit K bioaccessibility [22]. However, during digestion, the activity of digestive enzymes, along with changes in pH and ion concentrations, facilitates the breakdown of such complexes. This process may release K from macromolecular bindings, transforming it into a more bioaccessible form, supporting the observation of values above 100%. It has been noted that the metal-binding sites on peptides tend to become protonated under acidic conditions, which reduces their ability to interact with metal ions [23]. Although K is a dominant mineral in plants, its bioaccessibility data are relatively scarce. Scrob et al. [21] reported that K was the most bioaccessible mineral in dried fruits following in vitro digestion. In another study, K and Ca had the highest bioaccessibility levels in dried plants and infusions of *Achillea millefolium*, *Laurus nobilis*, and *Taraxacum sect*. *Ruderalia*, when assessed via membrane dialysis [24]. These findings align with the present study, underscoring the consistent bioavailability of K in plant-derived matrices.

Calcium concentrations in bee pollen peaked during the gastric digestion phase; however, a marked decline was observed in the intestinal phase compared to the oral phase, resulting in concentrations lower than those found in the undigested samples. As illustrated in Figure 2B, only approximately 25% of the total Ca content remained bioaccessible after the full digestion process, with the highest Ca bioaccessibility recorded in the BP4 multifloral bee pollen sample. These findings are consistent with previous studies on plant-based foods. For instance, Ca bioavailability has been reported to range between 11 and 32% in plant-based burgers [7], 17 and 62% in dried fruits [21], and 10 and 53% in Australian field pea, lentil, and sweet lupin [25]. The reduced bioaccessibility of Ca in the intestine has been attributed to several factors. Luz et al. [7] suggested that it results from interactions between minerals and other food matrix components such as proteins and lipids, rather than anti-nutritional compounds. On the other hand, Scrob et al. [21] emphasized the role of phytate, oxalate, and dietary fiber in reducing Ca availability by forming insoluble complexes. Additionally, Zhang et al. [25] noted that much of the Ca absorption occurs in the colon, beyond the typical scope of in vitro digestion models. Bee pollen’s complex matrix further complicates mineral bioaccessibility. With average compositions of 10–40% protein, 1–13% lipid, and 0.3–20% crude fiber [1], bee pollen also contains anti-nutritional factors like phytic acid, tannins, and protease inhibitors [26]. Moreover, the pollen’s structural barrier—primarily the exine layer made of sporopollenin—is chemically resistant and indigestible in the human gastrointestinal tract, which lacks the specific enzymes needed to degrade it [1,26]. Despite bee pollen’s rich nutritional profile, with over 200 bioactive compounds, these structural and compositional challenges significantly reduce bioaccessibility.

The mean Mg content in undigested bee pollen samples was 624 mg/kg, while the concentration in the digested samples averaged 595 mg/kg, indicating minimal loss during the digestion process. This suggests that the Mg present in bee pollen is relatively stable and remains largely accessible throughout the digestive stages. Its average bioaccessibility values were 77%, 135%, and 101% for oral, gastric, and intestinal digestion stages, respectively (Figure 2C). These findings imply that digestive conditions, particularly those in the stomach, may enhance the solubility and availability of Mg, likely due to the acidic environment facilitating its release.

According to the results, the mean Fe and Zn contents in bee pollen samples decreased substantially by approximately 18-fold and 15-fold, respectively, after complete in vitro digestion, compared to their levels in undigested samples. Among the samples, the highest Fe concentration (8.8 mg/kg) and bioaccessibility (26%) were observed in the BP4 multifloral bee pollen, while the *Salix* spp.-rich bee pollen of the BP2 sample exhibited the highest Zn content (33 mg/kg) and bioaccessibility (34%). Despite these peaks, the overall mean bioaccessibility levels of Fe and Zn were 9.5% and 7.5%, respectively, making them the least bioaccessible minerals analyzed in this study (Figure 2D,G). These findings align with previous reports, such as that by Scrob et al. [21], which also identified Fe and Zn as the least bioaccessible minerals in dried fruits post-digestion. Zhang et al. [25] similarly found low Fe and Zn bioaccessibility in legumes, with Fe absorption ranging from <0.1% to 13.4% and Zn absorption ranging from <0.1% to 24.2%, depending on the digestive stage and plant matrix. Sumczynski et al. [27] explained that Fe and Zn often precipitate as hydroxides or form complexes with phytic acid and other anti-nutrients, limiting their absorption. In plant materials, Fe is present in the non-heme form, which is more prone to oxidation and complexation [12]. Moreover, polyphenols are particularly inhibitors of Fe absorption due to their strong chelating ability, forming low-solubility complexes [25,27]. One potential explanation for the inconsistency in the Fe bioaccessibility trend across digestion stages and among bee pollen samples could be associated with the variability in the phenolic composition profiles of the pollen samples and their impact on the formation and stability of Fe–polyphenol complexes. As for Zn, phytic acid remains the primary inhibitor, especially in its pentaphosphate and hexaphosphate forms, which form stable complexes that hinder Zn uptake. Additionally, high Ca levels can exacerbate Zn malabsorption by enhancing the stability of phytic acid–Zn complexes [28].

The mean Mn contents measured during the oral, gastric, and intestinal digestion stages were 13, 29, and 7 mg/kg, respectively (Table 2), with corresponding bioaccessibility values of 44%, 86%, and 21% (Figure 2E). Notably, Mn bioaccessibility peaked during the gastric phase and declined sharply in the intestinal phase. This pattern aligns with findings reported by Scrob et al. [21] in dried fruits, as well as the findings of studies on ayurvedic plants [29] and legumes such as lentils, lupins, and field peas [25]. The enhanced Mn release in the gastric phase is attributed to the acidic pH and enzymatic activity of pepsin, which facilitates the breakdown of Mn–protein complexes and increases their solubility. However, during intestinal digestion, Mn availability is hindered primarily by the presence of phytic acid, which contains negatively charged phosphate groups that can strongly bind Mn, forming insoluble complexes and reducing its solubility [30]. Additionally, other dietary components such as fiber and certain minerals—namely Ca, K, and Mg—may competitively interact with Mn, further limiting its bioaccessibility in the gastrointestinal system [31].

The average copper values after oral and gastric digestion represent 20% of the initial content in the undigested sample, with a slight increase up to 24% during the intestinal phase. This behavior, more pronounced on *Nigella* spp. and *Helianthus annuus* samples, is similar to that observed for several dried fruits and is likely due to the enzymatic action and enhanced solubility of phytate–Cu complexes under alkaline conditions, as reported by Scrob et al. [21]. It is worth mentioning that although the release of copper increases from the gastric to the intestinal phase, the highest bioaccessibility in the intestinal phase is undoubtedly for magnesium and potassium, with values above 100%, compared to undigested bee pollen.

### 2.3. Assessment of Nutritional Value and Potential Health Risks

The contributions of bee pollen consumption to the recommended daily allowance (RDA) of minerals—K, Ca, Mg, Fe, Mn, Cu, and Zn—after in vitro digestion, along with non-carcinogenic health risk assessments via target hazard quotient (THQ) and the hazard index (HI) for Fe, Mn, Cu, and Zn, are presented in Table 3. Based on the bioaccessible mineral contents, the highest and lowest contributions to daily intake were observed for Mn and Ca, respectively. When assuming a daily intake of 40 g of bee pollen, the Mn contribution was 16% for women and 12% for men, whereas the Ca contribution was less than 1% of the RDA. The bioaccessible levels of K, Mg, and Cu were found to contribute between 6% and 10% to daily intake, while Fe and Zn contributed less than 5%. According to Regulation (EU) No. 1169/2011, a food must supply at least 15% of the Nutrient Reference Value (NRV) per serving to be labeled as a “source of” a nutrient, and at least 30% to qualify as being “high in” that nutrient [32]. Accordingly, none of the minerals in bee pollen, except for Mn in the context of women’s dietary requirements, reached the threshold to be qualified for such nutrition claims. Nevertheless, bee pollen may still provide dietary support, particularly for K and Cu.

From a physiological perspective, Mn is crucial for intracellular functions and acts as a cofactor for several enzymes, including manganese superoxide dismutase, glutamine synthetase, and arginase. Manganese deficiency has been linked to impaired growth and reproductive function [33]. Potassium, widely distributed across body tissues, plays key roles in intracellular osmolality, nerve function, hormone secretion, and the regulation of protein and glycogen synthesis [27,33]. Its deficiency can lead to hypokalemia. Copper, another essential trace element, is involved in critical enzymatic processes such as cytochrome c oxidase activity, collagen and keratin formation, and iron metabolism. Inadequate Cu intake can result in weakened arteries, liver dysfunction, and secondary anemia [33].

The nutritional properties of various bee pollen samples in terms of mineral content were also evaluated by Erdoğan et al. [4], who concluded that bee pollen is a nutritious food based on a long-term daily intake of 40 g. However, the mineral values reported in their study were considerably higher than those observed in the present study. This discrepancy is due to the fact that their evaluation was based on the total mineral content of raw bee pollen. While total mineral analysis provides useful compositional information, it does not reflect the fraction of minerals that are bioaccessible to the human body. In contrast, the present study employed the in vitro gastrointestinal digestion model to assess the bioaccessible mineral content to provide a detailed estimation of the nutritional value of bee pollen in the human diet.

According to the THQ and HI results for Fe, Mn, Cu, and Zn, all values were well below the established risk threshold of 1.0. The cumulative HI values were calculated as 0.086 for adult women (60 kg) and 0.073 for adult men (70 kg), indicating no appreciable non-carcinogenic health risk from the intake of these trace elements through 40 g/day bee pollen consumption. Similarly, Erdoğan et al. [34] reported that THQ values calculated from As, Cd, Cr, Cu, Fe, Mn, Ni, Pb, and Zn contents in undigested bee pollen at the same consumption level also remained below the risk limit. Their reported HI values, based on the raw bee pollen mineral content, ranged between 0.294 and 0.666, further supporting that bee pollen does not pose a non-carcinogenic health risk when consumed at typical dietary levels.

## 3. Materials and Methods

### 3.1. Materials

Bee pollen samples were collected from various regions of Türkiye during 2020, and their botanical origins and collection sites are detailed in Table 1. Pepsin and pancreatin enzymes used for the in vitro digestion process, as well as all analytical-grade reagents required for the analyses, were obtained from Merck (Darmstadt, Germany). Commercial standard solutions of the target minerals were sourced from PanReac AppliChem (Barcelona, Spain).

### 3.2. Analytical Methods

#### 3.2.1. Palynological Analysis of Bee Pollen Samples

Palynological analysis was conducted following the methods described by Barth et al. [35] and Mayda et al. [3]. Pollen preparations were examined using a Nikon Eclipse E400 microscope (Tokyo, Japan). For each sample, 500 pollen grains were counted in triplicate. Identification of pollen types was performed using reference slides and literature sources [3,35,36,37]. Pollen frequency categories were classified as follows, based on their relative abundance: dominant pollen (>45%), secondary pollen (16–45%), minor pollen (3–15%), and trace pollen (<3%) [38]. Botanical origins of bee pollen samples were evaluated according to ISO/FDIS 24382 [15].

#### 3.2.2. In Vitro Digestion

Analyses of oral, gastric, and intestinal phases of in vitro digestion for bee pollen samples were conducted based on the INFOGEST 2.0 standardized protocol, with minor modifications [39].

Simulated salivary fluid (SSF), gastric fluid (SGF), and intestinal fluid (SIF) were prepared following the procedures described in the protocol. For the oral phase, 2.0 g of bee pollen was mixed with SSF at a ratio of 1:4 (*w*/*v*), adjusted to pH 7.0, and incubated at 37 °C under orbital shaking at 100 rpm for 2 min. In the gastric phase, SGF (pH 3.0), containing 2000 U/mL of pepsin, was added to the oral bolus in a 1:1 (*v*/*v*) ratio and incubated at 37 °C under orbital shaking at 100 rpm for 120 min. For the intestinal phase, SIF (pH 7.0) supplemented with 100 U/mL of pancreatin and 10 mM of bile salts was added to the gastric chyme in a 1:1 (*v*/*v*) ratio and incubated under the same conditions for 120 min. The control/blank solutions for each digestion stage were prepared with deionized water instead of bee pollen. For analysis, at the end of the specific digestion phase, the samples were centrifuged at 15,000 rpm for 15 min at 4 °C. The supernatants were collected for mineral composition analysis. Mineral bioaccessibility (%) at each digestion stage was calculated independently, using the following Equation (1).(1)$$Bioaccessibility\left(\%\right)=\frac{Amount\;of\;mineral\;released\;through\;digestion}{Amount\;of\;mineral\;in\;undigested\;sample}\times 100$$

#### 3.2.3. Mineral Composition Analysis of Bee Pollen Samples

The mineral contents of bee pollen samples, including K, Ca, Mg, Fe, Mn, Cu, and Zn, were determined using an atomic absorption spectrometer (AAS; Perkin Elmer PinAAcle 900T, Waltham, MA, USA). For raw (undigested) bee pollen samples, standard microwave-assisted acid digestion was performed using the MARS 5 Digestion Microwave System (CEM Corporation, Matthews, NC, USA). Specifically, 0.25 g of bee pollen was placed into a PTFE digestion tube and mixed with 10 mL of 65% nitric acid (HNO_3_). Digestion was carried out at 200 °C for 45 min at 1600 W. After cooling, the digested samples were diluted to a final volume of 50 mL with deionized water [1,4,6]. For in vitro-digested and control samples, the gastrointestinal digestate or control sample was diluted with 65% HNO_3_ (1:4 *v*/*v*) and left at room temperature for 24 h prior to analysis [29,40]. While Fe and Zn were directly measured without chemical modification, for the accurate detection of K, Ca, Mg, Cu, and Mn, chemical modifiers—cesium chloride (CsCl), lanthanum chloride (LaCl_3_), and magnesium nitrate [Mg(NO_3_)_2_]—were added to the samples at a concentration of 1 g/L. Mineral quantification (mg/kg) was performed using calibration curves generated from standard solutions as follows: K (y = −121.93x^2^ + 3286.7x + 7355.4, R^2^: 0.999), Ca (y = 0.0613x + 0.0042, R^2^: 0.999), Mg (y = −0.0047x^2^ + 0.215x + 0.0015, R^2^: 0.999), Fe (y = −0.0118x^2^ + 0.1091x + 0.0101, R^2^: 0.993), Mn (y = −0.0115x^2^ + 0.2144x + 0.0026, R^2^: 0.999), Cu (y = −0.0116x^2^ + 0.1809x + 0.0008, R^2^: 1.000), and Zn (y = −0.0074x^2^ + 0.1278x + 0.0043, R^2^: 0.999) [1].

#### 3.2.4. Contribution of Minerals to the Recommended Daily Allowance, and Their Health Risk Assessment

The recommended daily allowance (RDA) values for the selected minerals are presented in Table 3 [4,41]. The percentage contribution of each mineral to the RDA (RDA%) for adult women and men was calculated based on an assumed daily consumption of 40 g of bee pollen [4] using the mineral concentrations measured after in vitro digestion. Furthermore, the estimated daily intake (EDI) of each mineral (mg/kg bw/day) was calculated for adult women (60 kg bw) and men (70 kg bw) using the following equation:(2)$$EDI=\frac{\left(Mineral\;content\;of\;bee\;pollen\;after\;digestion\right)\times (Ingestion\;rate)}{Bw}$$

Additionally, the potential non-carcinogenic health risks associated with bee pollen consumption were assessed by calculating the Target Hazard Quotient (THQ) and the Hazard Index (HI), using the following equations:(3)$$THQ=\frac{EFr\times ED\times FIR\times C}{RfD\times Bw\times AT}\times 10^{-3}$$(4)$$HI=\sum_{k=1}^{n}T{HQ}_{Fe}+{THQ}_{Mn}+{THQ}_{Cu}+{THQ}_{Zn}$$
where EFr, ED, FIR, C, and AT represent the exposure frequency (365 days/year), exposure duration (70 years), food ingestion rate (40 g/day), mineral concentration (mg/kg), and averaging time for non-carcinogens (ED × 365 days), respectively. In addition, the oral reference dose (RfD) for Fe, Mn, Cu, and Zn minerals were reported to be 0.70, 0.14, 0.04, 0.30 mg/kg/day, respectively [4,21,34].

#### 3.2.5. Statistical Analysis

All digestion procedures were performed in triplicate, and the results are expressed as mean ± standard error on a dry weight basis. Statistical analyses were conducted using the SAS software package (version 9.00; SAS Institute Inc., Cary, NC, USA). Analysis of variance (ANOVA) followed by Tukey’s multiple-range test was used to determine statistically significant differences among groups. A significance level of *p* < 0.05 was considered indicative of a statistically significant difference.

## 4. Conclusions

This study provides a comprehensive evaluation of the mineral composition and bioaccessibility of bee pollen samples from diverse botanical origins, along with their potential dietary contributions and associated health risks. Bee pollen was found to be rich in K, followed by Ca, Mg, and trace minerals, such as Fe, Mn, Cu, and Zn, with monofloral *Castanea sativa* pollen showing the highest overall mineral content, whereas *Nigella* spp. showed the lowest values for calcium, magnesium, and copper. Although revealing a high content of minerals, only a fraction of bee pollen became bioaccessible after simulated digestion, with the gastric phase showing the highest release for most minerals. Manganese exhibited the highest dietary contribution, particularly for women, whereas calcium had the lowest. While none of the minerals—except Mn for women—met the minimum threshold for nutritional claims, bee pollen may still serve as a supplementary source of K and Cu, besides Mn. Importantly, the health risk assessment confirmed that consuming 40 g/day of bee pollen poses no non-carcinogenic risk, and bee pollen can be considered a safe and beneficial dietary supplement in terms of mineral intake. The findings could help future studies related to the development of new bee pollen-based products or the fortification of food products with bee pollen. Additionally, evaluating other nutritional components present in the bee pollen matrix that could influence mineral availability would also be a valuable direction for further research.

## Figures and Tables

**Figure 1 molecules-30-02745-f001:**
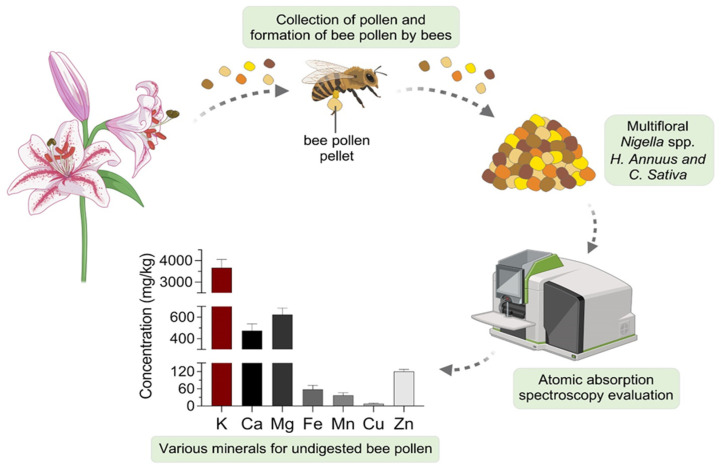
Collection of bee pollen by honeybees and the average mineral composition of undigested bee pollen samples.

**Figure 2 molecules-30-02745-f002:**
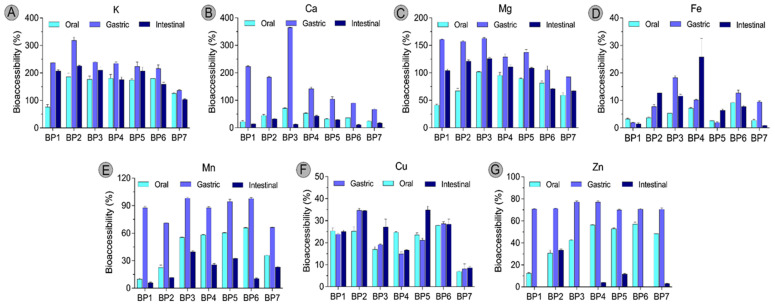
Changes in mineral bioaccessibility across oral, gastric, and intestinal digestion stages ((**A**): potassium; (**B**): calcium; (**C**): magnesium; (**D**): iron; (**E**): manganese; (**F**): copper; and (**G**): zinc). BP1 to BP7, sample number.

**Table 1 molecules-30-02745-t001:** Botanical and geographical origins of bee pollen samples.

Samples	Dominant Pollen (>%45)	Secondary Pollen (%16–45)	Minor Pollen (%3–15)	Trace Pollen (<%3)	Botanical Origin	Location
BP1	-	Fabaceae: 43 Rosaceae: 31 Asteraceae: 25	-	Lamiaceae: 1.2	Multifloral	Antalya
BP2	*Salix* spp.: 48	Fabaceae: 38	Cistaceae: 4.8 *Onobrychis* spp.: 4.8 Rosaceae: 4.8	-	Multifloral	Iğdır
BP3	*Nigella* spp.: 85	-	Apiaceae: 9.1 Rosaceae: 3.4	Brassicaceae: 1.1 Cistaceae: 1.1	*Nigella* spp.	Denizli
BP4	-	Brassicaceae: 29	*Trifolium* spp.: 15 Asteraceae: 12 *Campanula* spp.: 12 Papaveraceae: 10 Lamiaceae: 7.5 Cistaceae: 6.2 *Echium* spp.: 6.2	*Onobrychis* spp.: 1.2	Multifloral	Adıyaman
BP5	*Helianthus annuus*: 90	Chenopodiaceae: 6.9	-	*Cephalaria* spp.: 1.0 Fabaceae: 1.0 *Xanthium* spp.: 1.0	*Helianthus annuus*	Konya
BP6	-	Brassicaceae: 44 Cistaceae: 16	Rosaceae: 15 *Salix* spp.: 7.4 *Echium* spp.: 6.5 *Cichorium* spp.: 5.5 Papaveraceae: 4.6	Fabaceae: 0.9	Multifloral	Samsun
BP7	*Castanea sativa*: 83	-	*Lineria* spp.: 6.9	Cistaceae: 2.8 Brassicaceae: 2.1 Fabaceae: 2.1 Geraniaceae: 2.1 Rosaceae: 1.4	*Castanea sativa*	Yalova

**Table 2 molecules-30-02745-t002:** Concentrations of K, Ca, Mg, Fe, Mn, Cu, and Zn (mg/kg) minerals in bee pollen samples before and across in vitro digestion stages.

**Sample Name**	**K**				**Ca**			
	**Undigested**	**Oral**	**Gastric**	**Intestinal**	**Undigested**	**Oral**	**Gastric**	**Intestinal**
BP1	3410 ^Cc^ ± 10	2609 ^Ec^ ± 296	8092 ^BAa^ ± 66	7071 ^Ab^ ± 144	293 ^Eb^ ± 7	65 ^Dc^ ± 9	654 ^Ba^ ± 8	42 ^DCc^ ± 6
BP2	2420 ^Dc^ ± 38	4514 ^DCb^ ± 301	7715 ^Ba^ ± 294	5458 ^CBb^ ± 49	433 ^Db^ ± 8	196 ^CBc^ ± 14	799 ^Aa^ ± 8	142 ^Bc^ ± 4
BP3	3228 ^Cc^ ± 39	5728 ^DCBb^ ± 367	7744 ^Ba^ ± 33	6784 ^BAba^ ± 366	96 ^Fb^ ± 6	70 ^Dc^ ± 1	356 ^Ca^ ± 1	13 ^Dd^ ± 1
BP4	3304 ^Cc^ ± 40	5979 ^CBb^ ± 457	7772 ^Ba^ ± 160	5826 ^CBAb^ ± 288	431 ^Db^ ± 13	230 ^BAc^ ± 4	612 ^Ba^ ± 21	187 ^Ac^ ± 8
BP5	2476 ^Db^ ± 112	4330 ^Da^ ± 91	5548 ^Ca^ ± 398	5126 ^Ca^ ± 344	570 ^Ca^ ± 16	188 ^Cb^ ± 2	593 ^Ba^ ± 50	168 ^BAb^ ± 7
BP6	3963 ^Bc^ ± 93	7136 ^BAba^ ± 7	8582 ^BAa^ ± 524	6288 ^CBAb^ ± 263	667 ^Ba^ ± 10	246 ^Ac^ ± 1	602 ^Bb^ ± 6	72 ^Cd^ ± 10
BP7	6854 ^Ab^ ± 87	8643 ^Aa^ ± 225	9380 ^Aa^ ± 160	7131 ^Ab^ ± 146	835 ^Aa^ ± 21	203 ^CBc^ ± 6	564 ^Bb^ ± 22	145 ^Bc^ ± 6
**Sample Name**	**Mg**				**Fe**			
	**Undigested**	**Oral**	**Gastric**	**Intestinal**	**Undigested**	**Oral**	**Gastric**	**Intestinal**
BP1	623 ^Cb^ ± 3	258 ^Dc^ ± 9	1002 ^BAa^ ± 1	648 ^Bb^ ± 16	70 ^Ba^ ± 1	2.3 ^CBb^ ± 0.2	1.4 ^EDb^ ± 0.1	1.0 ^Bb^ ± 0.2
BP2	537 ^Dc^ ± 10	362 ^DCd^ ± 24	845 ^Ba^ ± 5	651 ^Bb^ ± 12	28 ^EDa^ ± 0	1.1 ^DCd^ ± 0.1	2.2 ^DCc^ ± 0.2	3.6 ^Bb^ ± 0
BP3	330 ^Fc^ ± 0	336 ^DCc^ ± 2	537 ^Da^ ± 5	416 ^Db^ ± 7	31 ^Da^ ± 1	1.7 ^DCBd^ ± 0.0	5.7 ^Bb^ ± 0.2	3.6 ^Bc^ ± 0.2
BP4	636 ^Cb^ ± 3	608 ^Bb^ ± 33	821 ^CBa^ ± 34	707 ^BAba^ ± 23	34 ^DCa^ ± 0	2.4 ^Bb^ ± 0.1	3.5 ^Cb^ ± 0.1	8.8 ^Ab^ ± 2.3
BP5	472 ^Ecb^ ± 3	423 ^Cc^ ± 4	649 ^DCa^ ± 23	513 ^Cb^ ± 7	38 ^Ca^ ± 1	1.0 ^Db^ ± 0.0	0.8 ^Eb^ ± 0.1	2.4 ^Bb^ ± 0.2
BP6	1075 ^Aba^ ± 12	879 ^Acb^ ± 38	1132 ^Aa^ ± 78	764 ^Ac^ ± 10	24 ^Ea^ ± 1	2.2 ^DCBb^ ± 0.0	3.0 ^Cb^ ± 0.2	1.8 ^Bb^ ± 0.1
BP7	698 ^Ba^ ± 2	415 ^Cb^ ± 31	647 ^DCa^ ± 2	469 ^DCb^ ± 7	178 ^Aa^ ± 1	4.9 ^Ac^ ± 0.6	16.9 ^Ab^ ± 0.6	1.6 ^Bc^ ± 0.2
**Sample name**	**Mn**				**Cu**			
	**Undigested**	**Oral**	**Gastric**	**Intestinal**	**Undigested**	**Oral**	**Gastric**	**Intestinal**
BP1	28.0 ^Ca^ ± 0.9	2.78 ^Ec^ ± 0.3	24.5 ^Cb^ ± 0.5	1.7 ^Ec^ ± 0.2	7.4 ^Ca^ ± 0.0	1.9 ^CBb^ ± 0.1	1.8 ^CBb^ ± 0.0	1.9 ^CBb^ ± 0.0
BP2	67.2 ^Ba^ ± 0.2	15.26 ^Bc^ ± 1.6	47.9 ^Bb^ ± 0.0	7.8 ^Bd^ ± 0.1	6.8 ^Da^ ± 0.0	1.7 ^Cc^ ± 0.1	2.4 ^BAb^ ± 0.1	2.4 ^BAb^ ± 0.0
BP3	19.5 ^Da^ ± 0.2	10.87 ^Cb^ ± 0.0	19.2 ^Da^ ± 0.2	7.8 ^Bc^ ± 0.2	6.1 ^Ea^ ± 0.0	1.0 ^Db^ ± 0.1	1.2 ^DCb^ ± 0.0	1.7 ^DCb^ ± 0.2
BP4	19.6 ^Da^ ± 0.2	11.42 ^Cc^ ± 0.1	17.1 ^Eb^ ± 0.2	5.0 ^Cd^ ± 0.3	8.8 ^Ba^ ± 0.1	2.2 ^BAb^ ± 0.0	1.3 ^DCc^ ± 0.1	1.5 ^DCc^ ± 0.0
BP5	10.2 ^Ea^ ± 0.3	6.18 ^EDb^ ± 0.1	9.7 ^Ga^ ± 0.3	3.3 ^Dc^ ± 0.1	7.7 ^Ca^ ± 0.2	1.8 ^CBc^ ± 0.1	1.6 ^Cc^ ± 0.1	2.7 ^Ab^ ± 0.1
BP6	14.6 ^EDa^ ± 0.3	9.64 ^DCb^ ± 0.1	14.3 ^Fa^ ± 0.2	1.6 ^Ec^ ± 0.1	8.5 ^Ba^ ± 0.0	2.4 ^Ab^ ± 0.0	2.4 ^Ab^ ± 0.1	2.4 ^BAb^ ± 0.2
BP7	101.4 ^Aa^ ± 2.3	36.39 ^Ac^ ± 0.4	67.4 ^Ab^ ± 0.4	23.4 ^Ad^ ± 0.6	11.6 ^Aa^ ± 0.1	0.8 ^Db^ ± 0.0	0.9 ^Db^ ± 0.3	1.0 ^Db^ ± 0.1
**Sample name**	**Zn**							
	**Undigested**	**Oral**	**Gastric**	**Intestinal**				
BP1	144.8 ^Ba^ ± 0.9	17.9 ^Dc^ ± 0.9	102.6 ^Bb^ ± 0.3	nd				
BP2	97.9 ^Ea^ ± 0.4	30.1 ^Cc^ ± 2.6	69.7 ^Eb^ ± 0.2	32.8 ^Ac^ ± 0.9				
BP3	119.6 ^Ca^ ± 0.0	51.1 ^Bc^ ± 0.1	92.5 ^Cb^ ± 1.0	nd				
BP4	100.7 ^Ea^ ± 0.1	56.7 ^Bc^ ± 0.2	77.3 ^Db^ ± 1.1	4.0 ^Cd^ ± 0.1				
BP5	106.1 ^Da^ ± 0.8	56.2 ^Bc^ ± 0.8	74.3 ^EDb^ ± 0.8	12.4 ^Bd^ ± 0.6				
BP6	89.7 ^Fa^ ± 0.4	51.3 ^Bc^ ± 1.5	63.4 ^Fb^ ± 0.5	nd				
BP7	178.6 ^Aa^ ± 0.9	86.5 ^Ac^ ± 0.2	126.0 ^Ab^ ± 1.9	5.5 ^Cd^ ± 0.5				

Differences between means within the same column are indicated by uppercase superscript letters (A, B, C, etc.), while lowercase superscript letters (a, b, c, etc.) denote differences among means within the same row, for each metal. “nd: not detected”.

**Table 3 molecules-30-02745-t003:** Nutritional and health risk assessment results of bee pollen consumption (40 g/day) after in vitro digestion.

Minerals	RDA Values (mg)		RDA%		EDI Values (mg/kg bw/day)		THQ Values		HI Values	
	Women	Men	Women	Men	Women	Men	Women	Men	Women	Men
K	2600	3400	9.6	7.4	4.165	3.570	-		0.086	0.073
Ca	1000	1000	0.4	0.4	0.071	0.061	-			
Mg	320	420	7.4	5.7	0.397	0.340	-			
Fe	18	8	0.6	1.4	0.002	0.002	0.003	0.002		
Mn	1.8	2.3	16.0	12.5	0.005	0.004	0.034	0.029		
Cu	0.9	0.9	8.3	8.3	0.001	0.001	0.031	0.027		
Zn	8	11	3.9	2.8	0.005	0.004	0.017	0.015		

## Data Availability

The data presented in this study are available on request from the corresponding authors.

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
