# Peer review of "Bee Pollen as a Dietary Mineral Source: In Vitro Bioaccessibility and Health Risk Evaluation"

_molecules, 2025, doi:10.3390/molecules30132745_

Round 1
Reviewer 1 Report
Comments and Suggestions for Authors
This manuscript describes a relevant and timely topic regarding the mineral composition, in vitro bioaccessibility, and health risk assessment of bee pollen. The research presents findings that are valuable both scientifically and nutritionally. The topic is important, especially considering the growing interest in functional foods and natural dietary supplements. The study contributes significantly to the literature by combining compositional analysis with bioaccessibility and toxicological risk evaluation—an approach not frequently applied to bee pollen.
The introduction is comprehensive and well-structured. It provides a clear rationale for the study by reviewing the nutritional components of bee pollen and emphasizing the importance of mineral bioaccessibility. The final paragraph effectively highlights the novelty and scientific contribution of the work.
The materials and methods section is well-detailed. It clearly describes the sampling procedure, the in vitro digestion protocol (based on INFOGEST 2.0), the analytical techniques (AAS), and the statistical analysis. The methodology is appropriate, reproducible, and adequately supported by previous literature.
The results are clearly presented and supported by figures and tables. The discussion compares this study to earlier research and explains differences in how well nutrients are absorbed based on the food's makeup, the digestion process, and how minerals interact. The authors successfully present their findings and engage in a discussion of relevant literature.
The conclusion effectively summarizes the major findings and emphasizes both the nutritional potential and safety of bee pollen consumption at recommended levels. It directs future research related to the bioavailability of other nutritional components.
Author Response
This manuscript describes a relevant and timely topic regarding the mineral composition, in vitro bioaccessibility, and health risk assessment of bee pollen. The research presents findings that are valuable both scientifically and nutritionally. The topic is important, especially considering the growing interest in functional foods and natural dietary supplements. The study contributes significantly to the literature by combining compositional analysis with bioaccessibility and toxicological risk evaluation—an approach not frequently applied to bee pollen.
Our response: We sincerely thank the reviewer for the evaluation and words of quality recognition on the work.
Reviewer 2 Report
Comments and Suggestions for Authors
The paper molecules-3694275 is devoted to the in vitro study of the bioavailability of bee pollen minerals. The paper makes a favorable impression. It is well-structured methodologically, the reliability of the results is beyond doubt, the work is novel and of practical significance. A comparison of our own results with the world level was conducted.
The following questions arose while reading.
- Please provide a link to the method of using a microwave digestion system described in section 3.2.3. Is this the author's development [1]? Please indicate how comparable it is with generally accepted methods.
- Please explain the results for Fe. Why does bioavailability increase for some types of honey at the stage of intestinal digestion, while it decreases for other types of honey?
Author Response
Reviewer #2:
The paper molecules-3694275 is devoted to the in vitro study of the bioavailability of bee pollen minerals. The paper makes a favorable impression. It is a well-structured methodologically, the reliability of the results is beyond doubt, and the work is novel and of practical significance. A comparison of our own results with the world level was conducted.
The following questions arose while reading.
Q1- Please provide a link to the method of using a microwave digestion system described in section 3.2.3. Is this the author's development [1]? Please indicate how comparable it is with generally accepted methods.
Our response: We thank the reviewer for this suggestion. In spectroscopic elemental analysis, acid digestion of the sample is a crucial step. The authors did not develop the method, but instead followed the most common procedure described in the literature for the preparation of pollen samples for mineral composition analysis. This procedure uses the microwave digestion method with acidic solvents, and can be easily found in many recent publications for pollen extraction before mineral composition analysis, such as:
- Aylanc, V., Falcão, S. I., & Vilas-Boas, M. (2023). Bee pollen and bee bread nutritional potential: Chemical composition and macronutrient digestibility under in vitro gastrointestinal system. Food Chemistry, 413, 135597.
- ErdoÄŸan, A., Åžeker, M. E., & Kahraman, S. D. (2023). Evaluation of environmental and nutritional aspects of bee pollen samples collected from East Black Sea region, Turkey, via elemental analysis by ICP-MS. Biological Trace Element Research, 201(3), 1488-1502.
- Valverde, S., Tapia, J. A., Pérez-Sanz, A., González-Porto, A. V., Higes, M., Lucena, J. J., ... & Bernal, J. (2023). Mineral composition of bee pollen and its relationship with botanical origin and harvesting period. Journal of Food Composition and Analysis, 119, 105235.
- Ecem Bayram, N. (2021). Vitamin, mineral, polyphenol, amino acid profile of bee pollen from Rhododendron ponticum (source of “mad honey”): nutritional and palynological approach. Journal of Food Measurement and Characterization, 15(3), 2659-2666.
Taking into consideration the comment, we improved the text and added some additional references for the method used in the present study (pag 15).
Q2: Please explain the results for Fe. Why does bioavailability increase for some types of honey at the stage of intestinal digestion, while it decreases for other types of honey?
Our response: Indeed, it is evident that the bioaccessibility of iron does not reveal a clear trend between digestion stages and samples. One potential explanation for this behavior can be linked to the phenolic profile variation between samples due to the differences in the botanical origin, and their ability or not to form complexes. To improve the manuscript, we incorporated this discussion into the manuscript as per the recommendation. (pag. 6)
Reviewer 3 Report
Comments and Suggestions for Authors
This research manuscript Mutlu et al presented “Bee Pollen as a Dietary Mineral Source: In Vitro Bioaccessibility and Health Risk Evaluation”. This manuscript is well written. However, needs developments to meet the journal quality. I recommend for major revision and my comments are:
- In the abstract, the most abundant details about bee pollen are missing.
- In the bee pollen, which digestive phase of copper is more bioaccessible?
- In the analytical evaluation of these abundant species, why atomic absorption spectroscopy techniques are used and their important information are missing.
- The significance of AAS for mineral resource details is missing, and it is recommended to include the advantages of AAS in comparison with other analytical methods.
- What were the main objectives of the study on bee pollen mineral content and bioaccessibility?
- Which mineral was found to be the most abundant in bee pollen samples, and which plant species' pollen showed the highest mineral concentrations?
- During which digestion phase were most minerals in bee pollen found to be most bioaccessible, and which mineral showed the highest release in the intestinal phase?
Author Response
Reviewer #3:
This research manuscript Mutlu et al presented “Bee Pollen as a Dietary Mineral Source: In Vitro Bioaccessibility and Health Risk Evaluation”. This manuscript is well written. However, needs developments to meet the journal quality. I recommend for major revision and my comments are:
This research manuscript Mutlu et al presented “Bee Pollen as a Dietary Mineral Source: In Vitro Bioaccessibility and Health Risk Evaluation”. This manuscript is well written. However, needs developments to meet the journal quality. I recommend for major revision and my comments are:
Q1: In the abstract, the most abundant details about bee pollen are missing.
Our response: We thank the reviewer for the comment. We improved the abstract by adding details on the bee pollen samples. (pag.1)
Q2: In the bee pollen, which digestive phase of copper is more bioaccessible?
Our response: The bioaccessibility of copper is very similar after the gastric and intestinal phases, with some exceptions for Nigella spp, Helianthus annuus samples, where the intestinal phase is significantly higher. Although we mentioned this aspect in the initial manuscript, we emphasize this behavior now in the abstract and in the discussion section, to make it more evident. (pag. 1, 4, and 6).
Q3: In the analytical evaluation of these abundant species, why atomic absorption spectroscopy techniques are used and their important information are missing. The significance of AAS for mineral resource details is missing, and it is recommended to include the advantages of AAS in comparison with other analytical methods.
Our response: We appreciate the reviewer's comment to improve the manuscript, and to do so, we added some details about the atomic absorption spectroscopy technique in the “introduction” section. (pag. 2)
Q4: What were the main objectives of the study on bee pollen mineral content and bioaccessibility?
Our response: Thanks for raising the question. We set four main objectives within our study:
(1) to determine the concentrations of macro- and trace minerals (K, Ca, Mg, Fe, Mn, Cu, and Zn) in bee pollen samples from various botanical and geographical origins;
(2) to evaluate the bioaccessibility of these minerals through simulated in vitro digestion using the standardized INFOGEST protocol
(3) to assess their potential contributions to the recommended daily intakes
(4) and to estimate associated health risks for adult men and women.
These objectives were stated at the end of the “Introduction” section. (pag.2)
Q5: Which mineral was found to be the most abundant in bee pollen samples, and which plant species' pollen showed the highest mineral concentrations?
Our response: While potassium was the most abundant mineral type in the bee pollen analyzed, the bee pollen sample with the highest mineral concentration was found to be Castanea sativa. This information is given in the “Abstract” and “Results and Discussion” sections, and added in the conclusions. (pag. 1,3, and 16).
Q6: During which digestion phase were most minerals in bee pollen found to be most bioaccessible, and which mineral showed the highest release in the intestinal phase?
Our response: The highest mineral bioaccessibility was found in the gastric phase, with some exceptions for copper and iron, where some samples showed higher bioaccessibility in the intestinal phase. The overall bioaccessibility order after simulated digestion was found to be K > Mg > Cu > Ca > Mn > Fe > Zn. Although Cu mineral was more bioaccessible in the intestinal digestion phase compared to the gastric phase, the minerals with the highest release in the intestinal phase were potassium and magnesium, with bioaccessibility above 100% when compared to undigested pollen. We improved the discussion section clarifying those points. (pag. 1 and 6)
Round 2
Reviewer 3 Report
Comments and Suggestions for Authors
Authors improved manuscript quality and recommed for pubcation.